# Using a Topical Formulation of Vitamin D for the Treatment of Vitiligo: A Systematic Review

**DOI:** 10.3390/cells12192387

**Published:** 2023-09-30

**Authors:** Khadeejeh Al-Smadi, Masood Ali, Seyed Ebrahim Alavi, Xuping Jin, Mohammad Imran, Vania R. Leite-Silva, Yousuf Mohammed

**Affiliations:** 1Frazer Institute, Faculty of Medicine, The University of Queensland, Brisbane, QLD 4102, Australia; k.alsmadi@uqconnect.edu.au (K.A.-S.); mohammad.imran@uq.edu.au (M.I.);; 2School of Pharmacy, The University of Queensland, Brisbane, QLD 4102, Australia; 3Departamento de Ciências Farmacêuticas, Instituto de Ciências Ambientais, Químicas e Farmacêuticas, Universidade Federal de São Paulo, UNIFESP-Diadema, São Paulo 09913-030, Brazil

**Keywords:** vitiligo, analogue, repigmentation, microneedling, single-blinded

## Abstract

Vitamin D is one significant prohormone substance in human organ systems. It is a steroidal hormone produced in the skin upon exposure to UVB rays. This paper presents a systematic review of the utilization of topical vitamin D, specifically cholecalciferol, calcipotriol, and tacalcitol, in the treatment of vitiligo. It considers the role of vitamin D in stimulating the synthesis of melanin and melanogenesis, which can help with the process of repigmentation. The inclusion of calcipotriol or tacalcitol in Narrowband Ultraviolet Phototherapy (NB-UVB) has shown the potential to enhance therapeutic outcomes for vitiligo. However, their effectiveness in combination with Psoralens Long Wave Ultraviolet Radiation (PUVA) and Monochromatic Excimer Light (MEL) treatment for vitiligo is limited. In contrast, combining topical corticosteroids with vitamin D analogues has demonstrated superior efficacy in treating vitiligo compared to using vitamin D analogues alone, while also providing the added benefit of reducing corticosteroid-related adverse effects. In addition, treating stable vitiligo with topical cholecalciferol and microneedling has shown success. Future studies are needed to ascertain an efficient method of administering vitamin D topically as an anti-vitiligo agent.

## 1. Introduction

Vitamin D is an essential fat-soluble steroidal hormone that can either be obtained from dietary sources, such as mushrooms or fish, or synthesized within the epidermal skin layers upon exposure to UVB rays [1,2]. Epidemiological research indicates that a significant proportion of the world’s population lacks adequate levels of vitamin D_3_, the active form of vitamin D [3,4]. Vitamin D deficiency affects over a billion individuals worldwide, making it increasingly prevalent [5,6,7]. Low blood vitamin D levels have been associated with an elevated risk of chronic diseases including cancer, hypertension, inflammation, and diabetes [8,9]. Furthermore, children born to pregnant women with low vitamin D levels have been found to exhibit autistic symptoms at the age of six, highlighting the potential benefit of vitamin D supplementation during pregnancy as a fundamental solution to this issue [10,11].

Vitiligo, an autoimmune pigmentary disorder characterized by the loss of functional melanocytes in the epidermis and infundibulum of the hair, manifests as clearly defined depigmented patches or macules [12,13]. Research [14,15] has shown that vitiligo impacts individuals of all skin colors and genders. Clinically, vitiligo can present in three different types: (a) unclassified vitiligo, which affects the majority or entirety of the body surface area, (b) segmental vitiligo, characterized by lesions following a dermatomal pattern, and (c) generalized vitiligo, the most common type, which displays bilateral and symmetrical distribution of the lesions [16,17,18]. Notably, individuals with vitiligo and other autoimmune diseases often exhibit low levels of vitamin D [19,20]. Vitamin D exerts a significant influence on the activity of keratinocytes and melanocytes through various mechanisms [21,22]. Firstly, it promotes the differentiation and maturation of keratinocytes, leading to the development of a well-structured epidermal barrier. This helps in maintaining the integrity of the skin and facilitates the repigmentation process in vitiligo [23]. Additionally, vitamin D has immunomodulatory effects, suppressing excessive immune responses that can contribute to melanocyte destruction in vitiligo. Furthermore, it enhances the production of melanin pigment within melanocytes, aiding in the repigmentation of depigmented skin patches [14]. Vitamin D also influences the release of various growth factors and cytokines that promote the survival and proliferation of melanocytes [24]. Elevated levels of thioredoxin have been associated with abnormal calcium uptake in the keratinocytes and melanocytes of vitiligo-affected skin, which can potentially inhibit melanogenesis by reducing tyrosinase activity [25,26,27]. While it is crucial for many people to consume foods fortified with vitamin D and obtaining a little sun exposure is important for establishing a healthy vitamin D level, certain populations, such as the elderly, obese individuals, dark-skinned populations, and breastfed newborns, may require dietary supplements to meet their vitamin D requirements [28,29,30]. Vitamin D and its metabolites play a vital role in various physiological functions [21]. For instance, they are essential in maintaining calcium homeostasis, promoting bone mineralization, and as a treatment option for a range of skin disorders, including psoriasis and vitiligo [8,14].

Several studies [14,31,32] have shown the therapeutic benefits of vitamin D supplementation in various experimental animal models, including those for allergic encephalomyelitis, collagen-induced arthritis, type 1 diabetes, inflammatory bowel disease, autoimmune thyroiditis, and systemic lupus erythematosus. Consequently, vitamin D supplements represent a promising treatment option for autoimmune conditions such as vitiligo [12]. Treatments for vitiligo usually include phototherapy as well as topical and oral immunomodulators such as corticosteroids and calcineurin inhibitors. The first-line therapies for vitiligo include topical corticosteroids (TCS) of moderate to high potency and calcineurin inhibitors (TCI), both of which suppress the cellular immune response [33,34]. Topical corticosteroids, commonly used for vitiligo, can lead to skin thinning and atrophy, especially with prolonged use. Some topical treatments, including calcineurin inhibitors, may cause skin irritation, burning, or itching, which can be uncomfortable for the patient. While phototherapy is effective, it may cause side effects such as redness, itching, and dry skin. Narrow-band UVB treatment, while safer than PUVA, can still lead to phototoxic reactions in some individuals [12,35]. These strategies have some efficacy in inducing repigmentation, although they do have certain drawbacks. This makes vitamin D therapy an alternative option for treating vitiligo.

The vitamin D family encompasses five molecules, with the two most prominent ones being vitamins D_3_ (cholecalciferol) and D_2_ (ergocalciferol) (Figure 1a and Figure 1b, respectively). Although both have beneficial effects on human health, they are obtained through different sources. Vitamin D_3_ is acquired from animal sources, such as fatty fish, liver, and eggs, whereas dietary vitamin D_2_ is commonly derived from plants, particularly mushrooms and yeast [36,37,38]. Both vitamin D_2_ and vitamin D_3_ exhibit the same affinity for the vitamin D receptor, indicating that neither form binds more strongly to the receptor [36,39]. Numerous studies [40,41] have demonstrated a significant difference in the effect on blood levels of circulating vitamin D between vitamin D_2_ and D_3_ supplementation. Vitamin D_3_ has been found to be more effective than vitamin D_2_ in increasing the body’s vitamin D and calcium levels, confirming that cholecalciferol supplementation is more efficient than ergocalciferol in improving vitamin D status [21]. Both vitamins D_2_ and D_3_ are inactive until they reach the liver, which modifies their chemical composition to create calcidiol, a molecule that the body uses to store vitamin D. In the kidneys, the active metabolite of the hormone, calcitriol (Figure 1c), is then formed from calcidiol (Figure 2) [36,42]. In dermatology, various derivatives (metabolites) of vitamin D, such as calcipotriol, calcitriol, tacalcitol, maxacalcitol, and hexafluoro-1,25 dihydroxy vitamin D_3_, are utilized [21,43]. Calcipotriol (Figure 1d), also known as calcipotriene, is commercially available under brand names, such as Dovonex^®^ and Diavonex^®^ (Leo Pharmaceutical Products, Ballerup, Denmark) It exhibits comparable efficacy to calcitriol due to its identical affinity for vitamin D_3_ receptors while causing less hypercalcemia than calcitriol [44,45]. Calcipotriol also exerts a significant impact on inflammatory mediators, the immune systems, and melanocytes, and it may enhance melanin formation by activating keratinocytes and melanocytes [46,47].

Tacalcitol (Figure 1e) is a synthetic vitamin D_3_ analogue that is comparable to calcitriol in terms of its affinity for vitamin D receptors and subsequent effects [44,48]. In a prospective open-label, left-right trial, the efficacy of two different vitamin D_3_ equivalents, calcipotriol and tacalcitol, in combination with NB-UVB phototherapy, was evaluated in patients with chronic stable plaque psoriasis. Thirty patients received NB-UVB phototherapy three times/week, along with daily application of tacalcitol to the target lesion on the left side and twice-daily application of calcipotriol to the right side. Both vitamin D_3_ analogues were deemed safe and effective, but calcipotriol exhibited greater efficacy, better absorption, faster onset of action, and more consistent treatment response [49]. Maxacalcitol (Figure 1f) shares the same topical mechanism as other vitamin D_3_ analogues, but it was reported to be 10 times more effective than calcipotriol and tacalcitol in reducing keratinocyte proliferation while being 60 times less calcemic than calcitriol [50,51].

Formulations are developed for active compounds like calcipotriol and tacalcitol to address their solubility and partitioning and to ensure appropriate dose delivery [52]. These analogues undergo specific modifications to enhance their affinity for vitamin D receptors while minimizing systemic side effects. Delivery enhancers play a crucial role in improving the skin permeability, solubility, and partitioning of the drug [53]. These enhancers include compounds that promote stratum corneum hydration and chemical penetration enhancers [54]. Strategies such as complexation, microencapsulation, or inclusion into certain delivery systems may be employed to enhance the solubility, bioavailability, and skin penetration of these compounds [55,56,57,58]. Micronutrients are commonly delivered using enhancers such as gums, oral tablets, buccal sprays, creams, gels, and ointments [59,60]. When designing topical skin formulations, it is essential to consider factors such as dosage calculations, delivery vehicles, drug distribution, and absorption control [61,62]. In medical facilities, ointments containing activated vitamin D_3_ derivatives are frequently employed for the treatment of psoriasis vulgaris, ichthyosis, vitiligo, and palmoplantar keratosis [44,63].

## 2. Melanin and Vitamin D Relation

Previtamin D_3_ is synthesized by solar exposure in the stratum Malpighii, which is composed of the stratum Basale and stratum Spinosum of the epidermis [64]. This region is principally where the epidermal 7-dehydrocholesterol reservoir is located (Figure 2) [65]. This thermally unstable previtamin undergoes temperature-dependent isomerization over the course of three days after formation to become vitamin D_3_ [66]. Plasma vitamin D binding protein preferentially transports vitamin D_3_ from the skin into circulation [67]. Increases in latitude or skin melanin levels inhibit the synthesis of previtamin D_3_ in the epidermis. Following a single complete body exposure to three minimum erythemal doses of UV radiation, serum vitamin D_3_ levels increased by almost 10 times, and serum 25-hydroxyvitamin-D levels doubled [68]. The most common cell types that make up the epidermis are keratinocytes and melanocytes. Melanocytes produce the pigment melanin, the skin color that distinguishes racial groups, mostly determined by the amount of melanin in the epidermis [69]. Most melanocytes are in the stratum Basale, the deepest layer of the epidermis, where the tyrosinase enzyme uses tyrosine to produce melanin [70]. Additionally, the long cytoplasmic processes of the melanocyte cells convey the melanin-containing pigment granules to other neighboring epidermal cells that are ascending towards the surface; therefore, each of the five epidermal strata contains melanin. The unique mechanism for the epidermal synthesis, storage, and continuous release of vitamin D into circulation prompted research into the potential therapeutic benefits of using the skin as a site for synthesizing and absorbing vitamin D_3_ metabolites [22,71,72]. Multiple studies have repeatedly shown that vitamin D_3_ can accelerate repigmentation by increasing tyrosinase activity and melanogenesis [14,24,73]. Moreover, vitamin D protects the epidermal melanin layer and restores melanocyte stability, firstly by regulating melanocyte activation, proliferation, migration, and pigmentation pathways and secondly by moderating T cell activation, which is probably connected with melanocyte disappearance in vitiligo [14]. Because melanin, the pigment that gives skin its color, filters UV rays, it is recognized that patients with hypovitaminosis D are at risk for developing skin pigmentation [14,74].

The existing research on the utilization of topical vitamin D, whether as a monotherapy or in combination, for the treatment of vitiligo is currently limited and inadequate [75,76,77]. Consequently, a pivotal research inquiry arises: “What is the efficacy and comparative effectiveness of topical vitamin D, encompassing various forms, such as calcipotriol and tacalcitol, in the management of vitiligo?”.

Aim/Hypothesis: This literature aims to conduct a comprehensive review of scholarly sources investigating the use of topical vitamin D in the treatment of vitiligo. The review will encompass various study designs, including randomized controlled trials and prospective comparative studies, to provide a thorough analysis of the current knowledge in this area. Additionally, studies examining different forms of topical vitamin D, such as calcipotriol and tacalcitol, will be included to evaluate their individual efficacy and comparative effectiveness. The ultimate goal of this systematic review is to gain a deeper understanding of the role of topical vitamin D in vitiligo treatment and offer valuable insights for clinicians, researchers, and patients seeking evidence-based approaches to managing vitiligo.

## 3. Methods

### 3.1. Search Strategy

The evaluation of vitamin D’s efficacy in topical treatment for vitiligo involved a literature search primarily conducted in February 2023. PubMed and Scopus databases were utilized for this purpose. The search keywords employed were “vitiligo” AND “vitamin D” OR “calciferol” AND “cholecalciferol” OR “ergocalciferol” OR “tacalcitol” OR “maxacalcitol” OR “calcipotriol” AND “cream” OR “ointment” OR “paste” OR “gel” OR “lotion” OR “solution” OR “foam” OR “suspension” OR “spray”. The search encompassed the period from 2003 to 2023 and included articles, books and book chapters, clinical trials, and randomized controlled trials. Additional filters were applied to narrow down the search results to the English language and exclude reviews or newspaper articles.

### 3.2. Study Screening and Data Extraction

The data obtained from the literature search were imported into EndNote 20 for organization and management. To determine the eligibility of the articles, two reviewers independently assessed the titles, abstracts, and full texts. A total of 327 articles were found on Scopus and 188 articles on PubMed, as shown in (Figure 3a). Both search engines yielded a similar distribution of articles. The included articles were evaluated based on the vitamin D analogues and types of topical vehicle formulation, as depicted in (Figure 3d,c). The titles and abstracts of the articles were screened for relevance, resulting in a total of 83 publications. These publications underwent a detailed review, and ultimately 27 studies were selected and summarized in (Table 1).

### 3.3. Inclusion and Exclusion Criteria

The selection of studies was carried out based on the following inclusion criteria: (1) study design: prospective or randomized controlled trials; (2) interventions: vitamin D derivatives, including cholecalciferol, ergocalciferol, calcipotriol, tacalcitol, and maxacalcitol; (3) topical vehicle formulation: cream, ointment, lotion, foam, solution, suspension, gel, spray, paste; (4) outcomes: evaluation of results based on the total number of vitiligo lesions on each participant’s body or each patch, as well as the degree of repigmentation on a quartile scale (<25% (poor), ≥25% (moderate), ≥50% (good), and ≥75% (excellent)). Exclusion criteria were (1) duplicate publications; (2) studies published outside the specified time frame (between 2003 and 2023); (3) studies that were not randomized controlled trials or prospective trials; (4) studies involving oral or injectable forms of vitamin D (this study focuses on topical administration); (5) studies without full-text availability. The selection process for inclusion and exclusion of articles is illustrated in the PRISMA flow diagram shown in (Figure 4).

## 4. Results

Based on the classification in (Figure 3b), calcipotriol had the highest proportion of publications (85%) in the Scopus search engine, followed by tacalcitol (33%) and cholecalciferol (16%). In the PubMed search engine, cholecalciferol (vitamin D_3_) had the largest proportion of publications (15%), followed by calcipotriol and tacalcitol, both with a percentage of 7%. Maxacalcitol and ergocalciferol had lower percentages (less than 10% in Scopus) or were not reported in publications in PubMed. In terms of topical vehicle formulation, ointment had the highest proportion of publications in both Scopus (83%) and PubMed (26%), which aligns with the lipophilic nature of vitamin D. Cream had the second highest proportion of publications for topical vehicle formulation (68% in Scopus and 15% in PubMed). Other formulations, such as solutions, suspensions, and gels, had lower proportions, but they were more popular in Scopus compared to PubMed, as shown in (Figure 3c). The findings from the 27 trials (*n* = 1198) summarized in (Table 1) revealed that calcipotriol accounted for nearly 70% of the topically used vitamin D analogues for vitiligo treatment. Tacalcitol and cholecalciferol contributed to 22% and 8% of the total, respectively. Ointment represented 77.5% of the vehicles utilized for the topical formulation, while cream and solution vehicles constituted 18.5% and 4% of the total, respectively. In terms of patient outcomes, a good response was achieved in 40% of the cases, while an excellent and poor response was observed in 26% each. A moderate response was reported in 18% of the cases.

## 5. Discussion

Several studies [84,90,103] have demonstrated the efficacy of calcipotriol in repigmenting vitiligo lesions, whether used alone or in combination with Psoralens Long Wave Ultraviolet Radiation (PUVA) or Narrowband Ultraviolet Phototherapy (NB-UVB). However, there have also been publications [104,105] that reported hyperpigmentation in psoriasis patients treated with phototherapy and calcipotriol. Baysal et al. [47] conducted an open prospective right-left comparative study involving 22 patients with generalized vitiligo of up to 9 months duration. Calcipotriol cream was applied twice a day on one side, while both sides received PUVA radiation twice weekly. The study reported a good response of 53.3% for the combination group and 53.11% for the PUVA alone group, with no statistically significant difference between the two groups (*p* = 0.980). In another study by Cherif et al. [83], a prospective right-left comparative study was performed to evaluate the efficacy of the combination of calcipotriol and PUVA in treating vitiligo. One side of the body received a twice-daily application of 0.005% calcipotriol ointment, while the other side remained untreated. PUVA radiation was administered three times per week on each side. The combination group achieved 69% poor to moderate repigmentation, compared to 52% for the PUVA alone group (*p* = 0.015), with a higher level of repigmentation observed on the calcipotriol-treated side. Parsad et al. [103] described the use of a topical ointment containing calcipotriol 50 µg/g in combination with PUVA for the treatment of vitiligo. They found that combining calcipotriol with PUVA therapy reduced the treatment duration by up to 18 months while maintaining high effectiveness and faster response. Vazquez et al. [101] conducted a prospective open study involving 10 patients with non-segmental vitiligo, using a combination of PUVA radiation and calcipotriene ointment. Their study reported a lower degree of repigmentation compared to the findings of Parsad et al. [103]. Overall, the combination therapy of calcipotriol and PUVA shows promise in the treatment of vitiligo, although the reported efficacy varies among different studies. Further research is needed to establish the optimal protocol, determine the long-term effectiveness, and assess the safety of this combination therapy in vitiligo treatment.

NB-UVB has been demonstrated to be an effective and safe treatment option for vitiligo [75]. Several prospective comparative studies have investigated the use of 0.005% calcipotriol ointment (apart from Arca et al., who used 0.05% calcipotriol) with or without NB-UVB on non-segmental symmetrical vitiligo patients. Kullavanijaya [91], Hartmann et al. [86], Arca et al. [81], and Khullar et al. [90] reported that (only) calcipotriol had no significant effect on the overall response rate, while NB-UVB monotherapy was found to be significantly effective in treating vitiligo (*p* < 0.05). However, conflicting results were observed in prospective comparison trials conducted by Ada et al. [79] and Goktas et al. [75]. Both studies treated NB-UVB to both sides of the body and applied calcipotriol cream twice daily to one side for non-segmental vitiligo lesions. Ada et al. [79] found no significant difference (*p* > 0.05) with the use of calcipotriol in combination with NB-UVB, while Goktas et al. [75] demonstrated that the combination resulted in a 51% higher early pigmentation and a good response compared to 39% with NB-UVB alone (*p* = 0.0006). In a study by Lotti et al. [94], 470 patients with segmental and non-segmental vitiligo and Fitzpatrick skin types I, II, III, and IV were enrolled in a 6-month treatment course. The effects of calcipotriol with NB-UVB were compared to those of tacrolimus, topical betamethasone (BM), and L-phenylalanine with NB-UVB. The combination treated with calcipotriol ointment 50 µg/g achieved good repigmentation with no side effects in 75.6% of cases, whereas 90.2% of the BM combination group experienced cutaneous atrophy as a side effect. In conclusion, NB-UVB therapy is effective as a monotherapy for vitiligo treatment. However, the role of calcipotriol in combination with NB-UVB remains debatable and may yield varying results. Further research is necessary to determine the optimal treatment approach and evaluate the potential side effects associated with different treatment combinations.

The xenon chloride excimer laser emits a precise 308 nm light wavelength in a focused form and is also available as an incoherent version called the excimer lamp [106]. In the treatment of vitiligo, Monochromatic Excimer Light (MEL) delivered through both lasers and lamps has demonstrated superior efficacy compared to NB-UVB and induces more cellular alteration than traditional UVB modalities [107]. Goldinger et al. [85] conducted a comparative right-left single-blinded trial with 10 individuals having non-segmental bilateral symmetrical lesions and Fitzpatrick skin types II and III to investigate the effect of calcipotriol on the effectiveness of MEL in treating vitiligo. One side received an application of 0.005% calcipotriol ointment, while MEL with a 308 nm light wavelength was administered to both sides. The study found no significant improvement in the efficacy of MEL after 15 months with the addition of calcipotriol (*p* > 0.05). In a recent study by Juntongjin [88], acral vitiligo was treated with MEL and calcipotriol, followed by monotherapy with either 0.005% calcipotriol ointment or clobetasol ointment. The study revealed no significant difference between the two treatments (*p* > 0.05), with partial repigmentation observed in 85% of lesions treated with calcipotriol and 77% of lesions treated with clobetasol.

The combination of topical calcipotriol and topical steroids has shown effectiveness in treating psoriasis while reducing the adverse effects of steroids. Kumaran et al. [92], Xing and Xu [102], Abdel and Ibrahim [78], and Ibrahim et al. [87] conducted studies on stable vitiligo lesions using a combination of BM 0.5 mg/g and calcipotriol ointment 50 g/g (equivalent to 0.05 mg/g and 0.005% concentration). These trials consistently reported successful repigmentation and a reduction in the adverse effects associated with BM. In a study by Akdeniz et al. [80], the effects of calcipotriol in combination with BM and NB-UVB were compared to calcipotriol combined with NB-UVB or NB-UVB alone on non-segmental vitiligo lesions. The researchers found that the combination therapy of calcipotriol and BM resulted in a repigmentation rate of 63.33%, compared to 60.67% for calcipotriol and 46.67% for NB-UVB alone. The only statistically significant difference was observed between the BM group and NB-UVB monotherapy (*p* = 0.0048). These findings indicate that the combination of calcipotriol and topical steroids, such as BM, can be effective in achieving repigmentation and minimizing adverse effects in the treatment of vitiligo lesions.

Fractionated lasers have emerged as an innovative technology for skin repair, creating microscopic therapeutic zones that facilitate the penetration of externally administered agents, thus improving efficacy without causing epidermal damage [108,109]. In a prospective, randomized, and comparative trial, Bakr et al. [82] treated 30 patients with stable non-segmental vitiligo using a CO_2_ laser, followed by three months of therapy with either calcipotriol 0.05% ointment, tacrolimus, or NB-UVB. Remarkable results were observed with all three treatments: 10% calcipotriol, 30% tacrolimus, and 40% NB-UVB phototherapy. Moreover, all combinations were found to be effective and safe, with NB-UVB therapy being the most efficacious. Following laser treatment, 70% of the patients experienced minor transient adverse effects such as pain, erythema, and crustations, which resolved spontaneously within a few days. In a comparative trial conducted by Gargoom et al. [84], the effects of 50 µg/g calcipotriol cream and ointment as monotherapy were examined in 18 patients with segmental and non-segmental vitiligo lesions for up to 6 months. The study revealed that ointment was more effective than cream, with 77.8% of patients showing improvement and 22.2% showing no response. Among those who exhibited improvement, 21.4% achieved excellent results, and 28.6% achieved a good response. These findings indicate the potential of fractionated lasers for enhancing the efficacy of externally administered agents in vitiligo treatment. The combination therapies involving calcipotriol, tacrolimus, and NB-UVB have demonstrated positive outcomes, while the use of calcipotriol ointment as monotherapy has also shown effectiveness in improving vitiligo lesions.

Several studies have indicated that formulations containing the same drug at the same concentration can exhibit varying rates of drug delivery through the skin due to the physicochemical properties of the vehicle used. Moreover, formulations employing the same ingredients can also result in different drug delivery rates based on the droplet size of the emulsion. In vitro, drug release tests provide valuable insights into evaluating the impact of formulation factors such as dose formulas, vehicle composition, and drug solubility in vehicles. These considerations are crucial when formulating topical preparations [110,111].

Formulation components have two primary effects on skin permeation: Firstly, they can modify the lamellar structure of intercellular lipids in the stratum corneum. These components have the potential to enhance or impede the permeation of the drug through the skin by altering the organization and packing of the intercellular lipids. This, in turn, influences the pathway and rate at which drug molecules can traverse the stratum corneum. Secondly, formulation components can impact the solubility properties of lipids. The solubility of both the drug and the vehicle in these lipids can have implications for drug permeation through the skin. The lipid composition of the stratum corneum is a significant determinant of skin permeability. The composition of lipids in the stratum corneum plays a vital role in determining skin permeability. Formulation components can modulate the solubility of the drug in these lipids, consequently influencing the drug’s ability to penetrate the skin [112,113,114].

Recent findings suggest that tacalcitol is more effective than calcipotriol when combined with NB-UVB phototherapy for the treatment of vitiligo [115]. In the study conducted by Katayama et al. [89], which was the first to report on the effectiveness of tacalcitol in treating vitiligo, 15 individuals with segmental and non-segmental vulgaris vitiligo were treated with tacalcitol ointment and sunlight exposure for up to three months. The study revealed a clinically good to excellent response in 40% of patients (*p* < 0.05). In contrast, Rodriguez et al. [98] conducted a randomized double-blind placebo-controlled investigation involving 80 patients with non-segmental vitiligo over a duration of 4 months. They applied topical tacalcitol ointment (4 µg/g) once a day at night and performed daily sunlight exposure for 30 min. The study found no significant reduction in lesion size or response of less than 25% response (*p* > 0.05), suggesting that the influence of tacalcitol on repigmentation is limited. To compare the effectiveness of NB-UVB phototherapy alone and in combination with tacalcitol for vitiligo treatment, Leone et al. [93] enrolled 32 adults with symmetrical and generalized vitiligo lesions and Fitzpatrick skin types II, III, and IV over 12 months. Patients were randomly selected on one side to apply tacalcitol ointment (4 µg/g) daily and received NB-UVB phototherapy twice a week on both sides. The combination treatment demonstrated a significantly shorter therapy duration (*p* = 0.0005) and higher repigmentation scores compared to the side treated by NB-UVB alone. Tacalcitol was found to enhance the effectiveness of treatment and accelerate the repigmentation process. In a study by Sahu et al. [99], the therapy duration was reduced to six months, and NB-UVB was administered three times a week. The combined group, treated with 4 µg/g tacalcitol ointment and NB-UVB, achieved a moderate response in 16.6% of cases, a good response in 53.3%, and excellent repigmentation in 30% of cases at the end of the treatment period (*p* < 0.001). These findings highlight the superior efficacy and accelerated repigmentation associated with the combination of tacalcitol and NB-UVB phototherapy in the treatment of vitiligo compared to monotherapy approaches.

In a randomized single-blind controlled experiment conducted by Lu-Yan et al. [95], 38 adults with symmetrical or close lesions were enrolled in an eight-week study. They used topical tacalcitol cream (2 µg/g) twice daily, in combination with 308-nm MEL once a week for each lesion. The combination treatment resulted in a significantly higher excellent repigmentation rate of 25.7% compared to 5.7% for excimer monotherapy (*p* < 0.05). The combination approach demonstrated early pigmentation with a lower total dosage. In a randomized single-blinded paired comparative study by Oh et al. [96], 20 patients with localized non-segmental vitiligo were treated for 16 weeks. Daily application of the ointment tacalcitol (20 µg/g) and twice-weekly administration of the excimer laser were used either as monotherapy or in combination. The study found no significant differences (*p* > 0.05) in repigmentation between the treatments. Tacalcitol, whether administered alone or in combination with MEL, had limited effects on repigmentation.

VD3, previously used as an oral treatment for psoriasis, gradually fell out of favor due to its associated hypercalcemic adverse effects. Consequently, limited research has been conducted on the topical delivery of VD3. In a prospective open trial by Onita [97], 27 adult patients with vulgaris vitiligo who had a poor clinical response to previous treatments such as topical corticosteroids and PUVA were treated with vitamin D_3_ ointment and PUVA phototherapy. The study demonstrated that approximately 48% of patients achieved moderate improvements of more than 30%. This combination therapy shows promise as an alternative treatment for vitiligo. In a recent prospective comparison study conducted by Salem et al. [100], 25 patients with stable segmental and non-segmental vitiligo (Fitzpatrick skin types II, III, and IV) were enrolled. Each patient received treatment for a minimum of two patches, with one patch treated using microneedle alone and the other patch treated with a combination of microneedle and topical cholecalciferol (0.5–2 mL solution). After 19 months of treatment, a 52% excellent to good response rate was observed with the combined therapy. Although there were no statistically significant differences in treatment response between the two types of vitiligo in the treated patches (*p* > 0.05), there was a difference in the pattern of pigmentation between the two groups (*p* = 0.013). These findings suggest that topical vitamin D_3_, when used in combination with microneedling, can be effective in treating stable vitiligo.

## 6. Conclusions

Vitamin D and its analogues, such as calcipotriol and tacalcitol, are commonly used topically for the treatment of pigmentation disorders. When used in conjunction with NB-UVB therapy, the addition of these derivatives has the potential to enhance the therapeutic outcomes for vitiligo. Tacalcitol has been shown to have stronger efficacy compared to calcipotriol in this regard. However, there is limited evidence supporting the use of vitamin D analogues to enhance the effectiveness of PUVA and MEL treatments for vitiligo, although some positive effects have been observed with the use of calcipotriol in a few studies. The combination of topical corticosteroids with vitamin D analogues has demonstrated greater efficacy in the treatment of vitiligo compared to using vitamin D analogues alone. Furthermore, incorporating vitamin D analogues in the treatment regimen can help mitigate the adverse effects associated with corticosteroids. However, additional research is needed to explore the optimal methods of topically administering vitamin D analogues as an anti-vitiligo agent. Further studies are necessary to investigate the specific protocols, dosages, and treatment durations to maximize the therapeutic benefits of vitamin D analogues in the management of vitiligo.

## Figures and Tables

**Figure 1 cells-12-02387-f001:**
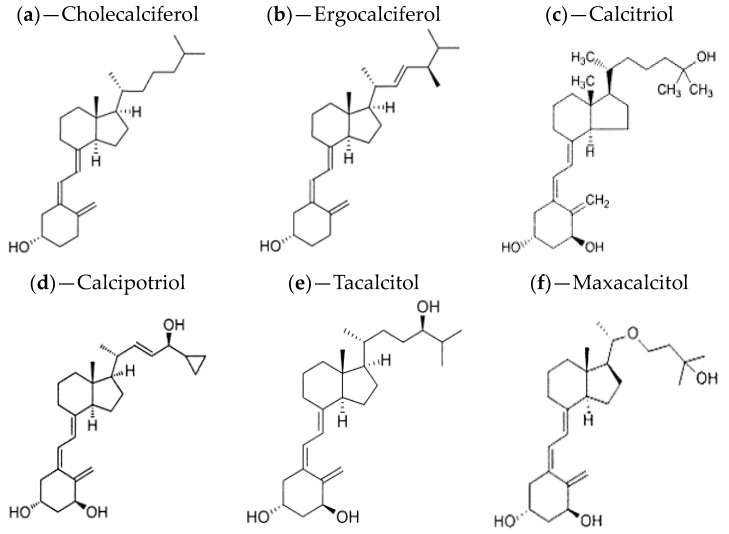
Chemical structures for types of vitamin D and its analogous [1].

**Figure 2 cells-12-02387-f002:**
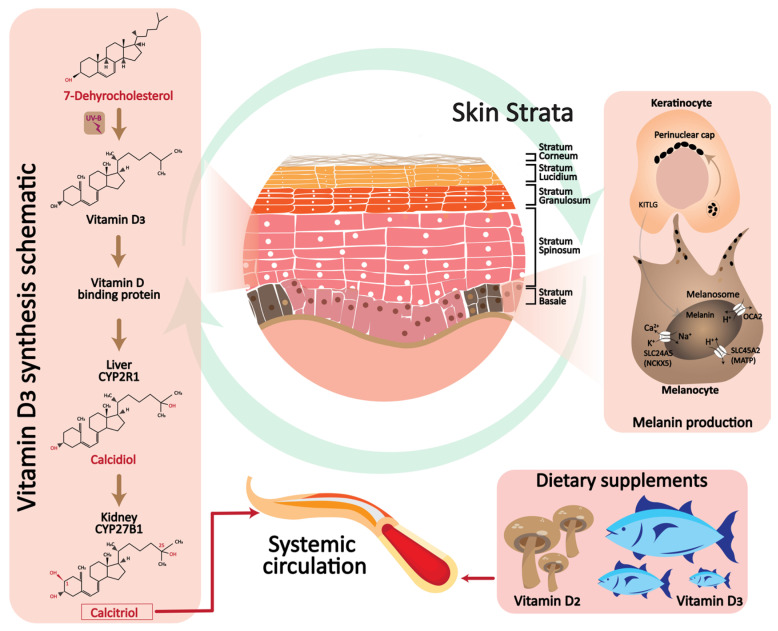
The relation between vitamin D and melanin. Abbreviations: CYP2R1, Cytochrome P450 Family 2 Subfamily R Member 1; CYP27B1, Cytochrome P450 Family 27 Subfamily B Member 1; KITLG, KIT-ligand; SLC24A5, solute carrier family 24 member 5; MATP, Membrane-associated transporter protein; OCA2, Oculocutaneous albinism type 2.

**Figure 3 cells-12-02387-f003:**
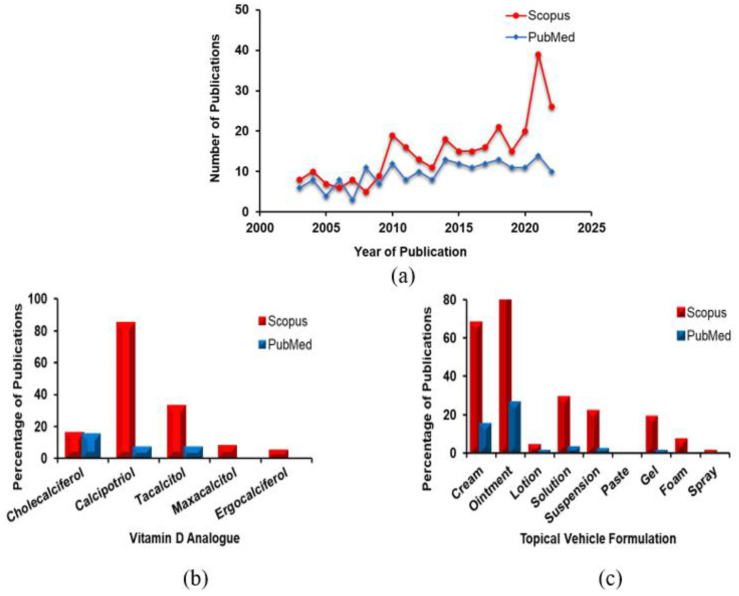
Analysis of search results. (**a**) The total number of journal articles published between 2003 and 2023 in Scopus *(n* = 327) and PubMed (*n* = 188) on the topic of vitamin D’s involvement in the treatment of vitiligo; (**b**) the subjective data indicated the proportions of journal articles published between 2003 and 2023 in Scopus and PubMed on vitamin D derivatives; and (**c**) types of topical vehicle formulations. Accessed database on 6 February 2023.

**Figure 4 cells-12-02387-f004:**
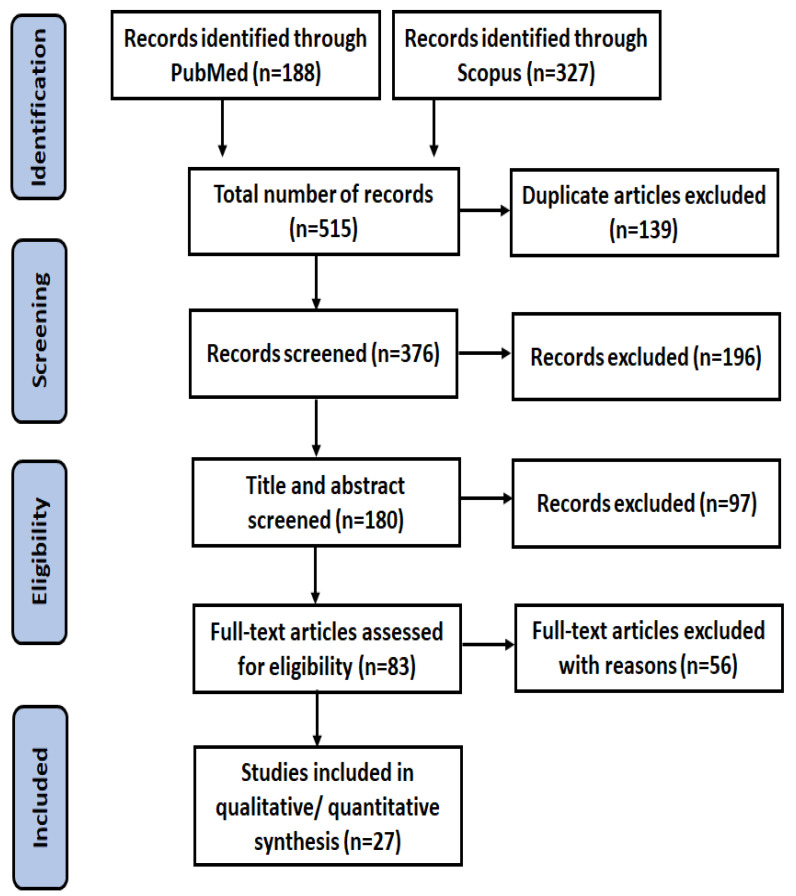
PRISMA diagram of the study identification.

**Table 1 cells-12-02387-t001:** Characteristics of included studies.

Source	Study Design	Types of Vitiligo	No. of Patients	Subtype	Treatment Duration	Intervention	Fitzpatrick Skin Type	Types of Topical Vit-D	Outcome
Abdel 2015 [78]	Randomized blinded comparative study	Non-segmental	44	Localized and stable	3 months	- Calcipotriol 50 µg/g with BM 0.5 mg/g vs. - Monochromatic excimer light (MEL)	NR	Ointment	- Combination achieved 19.4% excellent, 11.1% good, 33.3% moderate, and 16.6% poor repigmentation, while MEL achieved 16.6%, 27.7%, 13.8%, 19.4%, respectively
Ada 2005 [79]	Right-left comparison prospective single-blinded study	Non-segmental	20	Generalized vitiligo	12 months	- Calcipotriol 0.005% with NB-UVB	NR	Cream	- No significant difference for calcipotriol on NB phototherapy (poor)
Akdeniz [80] 2014	Randomized double-blind comparative study	Non-segmental	45	NR	6 months	- Calcipotriol with BM and NB-UVB vs. - Calcipotriol with NB-UVB vs. NB-UVB	NR	Ointment	- Combinations therapies of calcipotriol and BM achieved 63.33% to 60.67% for calcipotriol and 46.67% for NB alone (good)
Arca 2006 [81]	Comparative prospective study	Non-segmental	40	Stable	4 months	- Calcipotriol 0.05% with NB UVB vs. - NB-UVB	NR	Ointment	- Combination therapy achieved 45% of moderate repigmentation compared to 41.6% for NB alone - Calcipotriol did not add any advantages to NB
Bakr 2021 [82]	Prospective randomized comparative study	Non-segmental	30	Stable	3 months	- Calcipotriol 0.05% with CO_2_ laser vs. - Tacrolimus with CO_2_ laser vs. - NB-UVB with CO_2_ laser	NR	Ointment	- Excellent achievements for calcipotriol, tacrolimus, and NB was 10%, 30%, and 40%, respectively - All combinations are effective and safe - The NB group is the most effective one
Baysal 2003 [47]	Prospective right/left comparative, open study	Non-segmental	22	Generalized	9 months	- Calcipotriol with PUVA vs. - PUVA	NR	Cream	- The average response for the combination was (good) 53.3% and 53.11% for PUVA alone - No significant difference
Cherif 2003 [83]	Prospective study	Non-segmental	23	Bilateral symmetrical	15 weeks	- Calcipotriol 0.005% with PUVA vs. - PUVA	IV, V	Ointment	- Combination achieved 69% of poor to moderate repigmentation compared to 52% for PUVA alone
Gargoom 2004 [84]	Prospective right/left comparison study	Segmental and non-segmental	18	Focal/mucosal	4–6 months	- Calcipotriol 50 µg/g	NR	Cream/Ointment	- 77.8% showed improvement while 22.2% had no response. - From the improvement cases 21.4% achieved excellent and 28.6% good - Ointment is more effective than the cream
Goktas 2006 [75]	Prospective right/left comparison study	Non-segmental	24	Generalized	6 months	- Calcipotriol 0.05 mg/g with NB-UVB vs. - NB-UVB	II, III	Cream	- Early pigmentation/good (combination 51% compared to 39% for NB only)
Goldinger 2007 [85]	Prospective right/left comparative single-blinded trial	Non-segmental	10	Bilateral symmetrical	15 months	- Calcipotriol 0.005% with MEL vs. - MEL	II, III, IV	Ointment	- 88.2% showed evidence/of good repigmentation for both. - Effectiveness of MEL is not considerably increased with calcipotriol
Hartmann 2005 [86]	Prospective right/left comparative study	Non-segmental	10	Symmetrical (vulgaris/acrofacial /localized)	12 months	- Calcipotriol 0.005% with NB-UVB vs. - NB-UVB vs.	II, III	Ointment	- NB is effective/good in treating vitiligo - No therapeutic effect for calcipotriol in the combination (poor)
Ibrahim 2019 [87]	Prospective right/left comparative study	Segmental and non-segmental	25	Stable and symmetrical, focal and generalized	6 months	- Calcipotriol 0.05 mg/g with BM 0.5 mg vs. - Tacrolimus (both received derma pen microneedles	III, IV	Ointment	60% showed excellent improvement for the calcipotriol combination compared to 32% for tacrolimus
Juntongjin 2021 [88]	Prospective randomized double-blind comparative study	Non-segmental	13	Acral vitiligo	24 weeks	- Calcipotriol 0.005% with MEL vs. - Clobetasol with MEL	IV	Ointment	- 85% and 77% showed partial repigmentation for calcipotriol and clobetasol, respectively (poor). - No significant difference between treatments
Katayama 2003 [89]	Prospective uncontrolled open trial	Segmental and non-segmental	15	Vulgaris vitiligo	3 months	- Tacalcitol with solar irradiation	NR	Ointment	40% of patients responded in a good to excellent way clinically
Khullar 2015 [90]	Prospective right-left comparative study	Non-segmental	27	Generalized vitiligo (vulgaris and acrofacial	24 weeks	- Calcipotriol 0.005% with NB-UVB vs. - NB-UVB	III, IV, V	Ointment	51.4% reduction in the lesions for NB and 49% for combination (good repigmentation)
Kullavanija-ya 2004 [91]	Prospective open bilateral comparison study	Non-segmental	20	Symmetrical vitiligo	15 months	- Calcipotriene 0.005% with NB-UVB vs. - NB-UVB/placebo	NR	Ointment	- Calcipotriene did not affect the overall response rate (poor) - Excellent repigmentation 52.9% for the combination
Kumaran 2006 [92]	Randomized trial	Non-segmental	49	Localized vitiligo	3 months	- Calcipotriol 0.005% vs. - BM vs. - Calcipotriol with BM	NR	Ointment	- Marked/good repigmentation was 6.7%, 13.3%, and 26.7% for calcipotriol, BM, and combination, respectively - The combination reduced the negative effects of BM
Leone 2006 [93]	Prospective randomized open-label study	Non-segmental	32	Generalized and symmetrical	12 months	- Tacalcitol 4 µg/g with NB-UVB vs. - NB-UVB	II, III, IV	Ointment	- 35.9% good repigmentation for both - Tacalcitol improved the treatment and repigmentation occurred earlier
Lotti 2008 [94]	Prospective open study	Segmental and non-segmental	470	Vulgaris vitiligo	6 months	- NB-UVB vs. - NB-UVB with - Tacrolimus/Pimecrolimus/BM/Calcipotriol 50 µg/g - /L-phenylalanine	I, II, III, IV	Ointment	- 75.6% excellent repigmentation achieved for calcipotriol combination. - BM is the most effective combination
Lu-Yan 2006 [95]	Randomized single-blind within a patient-controlled trial	Segmental and non-segmental	38	Symmetrical, localized, vulgaris	8 weeks	- Tacalcitol 2 µg/g with MEL vs. - MEL	NR	Cream	25.7% excellent repigmentation in the combination (early pigmentation with lower total dosage) to 5.7% for excimer monotherapy
Oh 2011 [96]	Prospective randomized single-blinded paired comparative study	Non-segmental	20	Localized	16 weeks	- Tacalcitol 20 µg/g vs. - MEL vs. - Tacalcitol with MEL	NR	Ointment	- Limited/poor effects on repigmentation for tacalcitol either monotherapy or in combination
Onita 2004 [97]	Prospective open trial	Segmental and non-segmental	27	Vulgaris vitiligo	NR	- Vitamin D_3_ (Cholecalciferol) with PUVA	NR	Ointment	48% of patients improved > 30%(moderate)
Rodriguez 2009 [98]	Randomized double-blind placebo-controlled study	Non-segmental	80	NR	4 months	- Tacalcitol 4 µg/g with sunlight exposure	NR	Ointment	There was no reduction of the size of the lesion < 25% (poor repigmentation)
Sahu 2016 [99]	Prospective open label right/left intraindividual trial	Non-segmental	30	Symmetrical vitiligo	24 weeks	- Tacalcitol 4 µg/g with NB-UVB vs. - NB-UVB	NR	Ointment	16.6% had moderate. 53.3% good 30% excellent repigmentation at the end of therapy
Salem 2023 [100]	Prospective comparative study	Segmental and non-segmental	25	Stable vitiligo	19 months	- Vitamin D_3_ (Cholecalciferol) 0.5–2 mL with microneedles vs. - Microneedles	II, III, IV	Solution	52% excellent to good response in the combination
Vazquez 2003 [101]	Prospective open pilot study	Non-segmental	10	NR	NR	- Calcipotriene with PUVA	NR	Ointment	- The degree of repigmentation (moderate) was lower than that of (Parsad 1998)
Xing 2012 [102]	Prospective open uncontrolled trial	Segmental and non-segmental	31	Focal or generalized	12 weeks	- Calcipotriol 0.005% with BM	NR	Ointment	- 9.7% had an excellent response. - The combination was effective and tolerated

Abbreviations: NR, Not Reported; PUVA, Psoralens Long Wave Ultraviolet Radiation; NB-UVB, Narrowband Ultraviolet Phototherapy; MEL, Monochromatic excimer light; BM, Betamethasone.

## Data Availability

Not applicable.

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
