# Peer review of "Using a Topical Formulation of Vitamin D for the Treatment of Vitiligo: A Systematic Review"

_cells, 2023, doi:10.3390/cells12192387_

Round 1

Reviewer 1 Report

In the manuscript titled "Using a topical formulation of Vitamin D for the treatment of Vitiligo: A systematic review", the authors systematically reviewed the application of topical vitamin D, particularly calcipotriol, calcitriol, and tacalcitol, in the treatment of vitiligo, as well as the potential of adding calcipotriol or tacalcitol to NB-UVB phototherapy to enhance therapeutic outcomes. The manuscript doesn’t address the pros and cons of current therapeutic approaches for vitiligo, as well as the current clinical utilization status of vitamin D in vitiligo treatment. Additionally, there are issues regarding incomplete graphical details.

1.        The authors have extensively summarized various treatment regimens of vitamin D for vitiligo, covering aspects such as mode of administration, duration, and prognosis. However, there is a limited depiction of cellular and molecular biology aspects, which should be aligned with the theme of the journal.

2.        In the Introduction, in addition to providing a basic description of the relationship between vitamin D and vitiligo, it would be beneficial to introduce the current treatment approaches for vitiligo and highlight the limitations of these approaches, emphasizing the advantages of vitamin D therapy.

3.        In Section 3.3 of the Methods, a more detailed description and visual representation, such as a schematic diagram, should be provided for the outcome evaluation criteria in the inclusion criteria.

4.        In the discussion section, in addition to the treatment of vitiligo, authors can also discuss a variety of vitamin D treatments in other melanin and the prospect of the related disease treatment.

5.        In Figure 1, the figure is incomplete and needs to be modified.

Can be improved.

Author Response

Response to Reviewers

We would like to express our gratitude to the editor and reviewers for their time and effort in reading and commenting on our paper. The constructive comments have been considered, and the manuscript has been updated accordingly. The responses to the reviewers' comments are listed below, and they are highlighted throughout the manuscript.

Reviewer 1

  • Comment 1- The authors have extensively summarized various treatment regimens of vitamin D for vitiligo, covering aspects such as mode of administration, duration, and prognosis. However, there is a limited depiction of cellular and molecular biology aspects, which should be aligned with the theme of the journal.
  • Response: We appreciate your feedback and would like to draw your attention to the (section 2 Melanin and Vitamin-D Relation) Lines 154-191 of the article where we extensively covered the relationship between melanin and vitamin D. Also, the mechanism for the epidermal synthesis involves the transportation of vitamin D, which is produced in the keratinocytes of the epidermis, to the liver, where it undergoes hydroxylation to 25(OH)D, that acts as a prohormone and is activated in the kidney to form calcitriol.

Comment 2-  In the Introduction, in addition to providing a basic description of the relationship between vitamin D and vitiligo, it would be beneficial to introduce the current treatment approaches for vitiligo and highlight the limitations of these approaches, emphasizing the advantages of vitamin D therapy.

  • Response: We would like to thank the reviewer for the valuable comment, we have incorporated the suggested information about current treatment approaches for vitiligo and highlight the limitations of these approaches: Below details have been added to the introduction of manuscript and the reference list.
  • Lines 83-95:

Treatments for vitiligo usually include phototherapy as well as topical and oral immunomodulators such as corticosteroids and calcineurin inhibitors. The first-line therapies for vitiligo include topical corticosteroids (TCS) of moderate to high potency and calcineurin inhibitors (TCI), both of which suppress the cellular immune response [33, 34]. Topical corticosteroids, commonly used for vitiligo, can lead to skin thinning and atrophy, especially with prolonged use. Some topical treatments, including calcineurin inhibitors, may cause skin irritation, burning, or itching, which can be uncomfortable for the patient. While phototherapy is effective, it may cause side effects such as redness, itching, and dry skin. Narrow-band UVB treatment, while safer than PUVA, can still lead to phototoxic reactions in some individuals [12, 35]. These strategies have some efficacy in inducing repigmentation, although they do have certain drawbacks. This makes vitamin D therapy an alternative option for treating vitiligo.

  • References Lines 557-562:
  1. Bilal, A. and I. Anwar, Guidelines for the management of vitiligo. Journal of Pakistan Association of Dermatologists, 2014. 24(1): p. 68-78.
  2. Bouceiro Mendes, R., M. Alpalhão, and P. Filipe, UVB phototherapy in the treatment of vitiligo: State of the art and clinical perspectives. Photodermatology, Photoimmunology & Photomedicine, 2022. 38(3): p. 215-223.
  3. Bikle, D.D., Vitamin D metabolism, mechanism of action, and clinical applications. Chemistry & biology, 2014. 21(3): p. 319-329.

Comment 3-   In Section 3.3 of the Methods, a more detailed description and visual representation, such as a schematic diagram, should be provided for the outcome evaluation criteria in the inclusion criteria.

  • Response: We appreciate the reviewer's insightful feedback, regarding the 3.3 of the Methods, a more detailed description and visual representation, such as a schematic diagram: We have incorporated PRISMA flow diagram in (section 3.3 Lines 237-254 ) “Inclusion and Exclusion Criteria (Figure 4. PRISMA diagram of the study identification).

Comment 4.    In the discussion section, in addition to the treatment of vitiligo, authors can also discuss a variety of vitamin D treatments in other melanin and the prospect of the related disease treatment.

  • Response: We value the reviewer’s perceptive comment. Unfortunately, as the relationship between topical vitamin D and the treatment of vitiligo have been covered at the introduction, methodology, and results sections in this review we have to discuss according to these sections.

Comment 5.    In Figure 1, the figure is incomplete and needs to be modified.

  • Response: We value the reviewer's accurate suggestion and the figure 1 become Figure 2 and it has been modified. (Line 187)

Reviewer 2 Report

In the manuscript entitled “Using a topical formulation of Vitamin D for the treatment of Vitiligo: A systematic review”, Khadeejeh AL-Smadi et al. described a systematic review of the utilization of topical vitamin D, specifically cholecalciferol, calcipotriol, and tacalcitol in the treatment of vitiligo. They summarized a lot of relevant literature, including randomized controlled trials and prospective comparative studies, and have thoroughly analyzed the current knowledge in this area. However, some mistakes need to be corrected.

1.     Table 1 Characteristics of included studies:

Type of Vitiligo to Types of Vitiligo, No of Patients to No. of Patients, bilateral symmetrical to Bilateral symmetrical, Intervention, “1. 2.” should be deleted.

2.     References:

1)     Norman, A., Vitamin D. 2012: Elsevier. Book citations are missing pages.

2)     Agostini, D. and S. Donati Zeppa, Vitamin D, Diet and Musculoskeletal Health. 2023, MDPI. p. 2902. Lack of journal name “Nutrients”.

3)     The title font format is inconsistent, such as 7, 11, 12, 20, 25.

Author Response

In the manuscript entitled “Using a topical formulation of Vitamin D for the treatment of Vitiligo: A systematic review”, Khadeejeh AL-Smadi et al. described a systematic review of the utilization of topical vitamin D, specifically cholecalciferol, calcipotriol, and tacalcitol in the treatment of vitiligo. They summarized a lot of relevant literature, including randomized controlled trials and prospective comparative studies, and have thoroughly analyzed the current knowledge in this area. However, some mistakes need to be corrected.

Comment 1.     Table 1 Characteristics of included studies:

Type of Vitiligo to Types of Vitiligo, No of Patients to No. of Patients, bilateral symmetrical to Bilateral symmetrical, Intervention, “1. 2.” should be deleted.

  • We appreciate the reviewer's insightful feedback. All mistakes have been corrected according to reviewer suggestions. (Line 255)

Comment 2.     References:

1)     Norman, A., Vitamin D. 2012: Elsevier. Book citations are missing pages.

2)     Agostini, D. and S. Donati Zeppa, Vitamin D, Diet and Musculoskeletal Health. 2023, MDPI. p. 2902. Lack of journal name “Nutrients”.

  • We would like to thank the reviewer for the accurate comments. All the references have been modified according to his suggestions. (Lines 499-500)

Comment 3)     The title font format is inconsistent, such as 7, 11, 12, 20, 25.

  • We would like to thank the reviewer for the valuable comments. All font formats have been corrected.

Reviewer 3 Report

The manuscript summarizes the current state of knowledge on application of Vitamin D in the treatment of vitiligo. The paper is quite clearly written and the search strategy is seem to be appropriate.

However, I have some suggestions for Authors:

Add the figure with the chemical structures of forms of vitamin D and synthetic analogues

„Vitamin D exerts a significant influence on the activity of keratinocytes and melanocytes through various mechanisms” – write a few words about the mechanisms

Figures should be placed near first mention in the text as it is possible.

Figure 1 is not complete (see: melatonin produc). Furthermore, all abbreviation used in Figure should be explained in figure legend.

Table 1. all abbreviation used should be explained in footnote below the table.

Correct: „, w. Parsad”

Author Response

The manuscript summarizes the current state of knowledge on the application of Vitamin D in the treatment of vitiligo. The paper is quite clearly written, and the search strategy seems to be appropriate.

However, I have some suggestions for Authors:

Comment- Add the figure with the chemical structures of forms of vitamin D and synthetic analogues.

  • Thank you for your critical feedback, we have expanded the suggested section in the introduction part: figure with the chemical structures of forms of vitamin D and synthetic analogues has been added.

Line 153 Figure 1. Chemical structures for types of vitamin D and its analogous.

a- Cholecalciferol

b- Ergocalciferol

c- Calcitriol

d- Calcipotriol

e- Tacalcitol

f- Maxacalcitol

Reference Line 499

Norman, A., Vitamin D. 2012: Elsevier. p.1-101.

Comment:  „Vitamin D exerts a significant influence on the activity of keratinocytes and melanocytes through various mechanisms” – write a few words about the mechanisms.

  • We value the reviewer’s perceptive comment. We have incorporated the suggested information regarding mechanisms in the introduction. Below details have been added to the introduction of manuscript and the reference list.
  • Lines 56-65:

Firstly, it promotes the differentiation and maturation of keratinocytes, leading to the development of a well-structured epidermal barrier. This helps in maintaining the integrity of the skin and facilitates the repigmentation process in vitiligo [5]. Additionally, vitamin D has immunomodulatory effects, suppressing excessive immune responses that can contribute to melanocyte destruction in vitiligo. Furthermore, it enhances the production of melanin pigment within melanocytes, aiding in the repigmentation of depigmented skin patches [6]. Vitamin D also influences the release of various growth factors and cytokines that promote the survival and proliferation of melanocytes [7].

  • References Lines 538-541
  1. Hawker, N.P., et al., Regulation of human epidermal keratinocyte differentiation by the vitamin D receptor and its coactivators DRIP205, SRC2, and SRC3. Journal of investigative dermatology, 2007. 127(4): p. 874-880.
  2. Birlea, S.A., G.E. Costin, and D.A. Norris, Cellular and molecular mechanisms involved in the action of vitamin D analogs targeting vitiligo depigmentation. Current drug targets, 2008. 9(4): p. 345-359.

Comment:  Figure 1 is not complete (see: melatonin product). Furthermore, all abbreviations used in the Figure should be explained in the figure legend. 

  • We would like to thank the reviewer for the accurate comments. The figure 1 become Figure 2 and it has been modified. (Line 187)

Comment:  Table 1. All abbreviations used should be explained in a footnote below the table. 

Correct: „, w. Parsad”.  

  • We appreciate the reviewer's insightful feedback. All abbreviations have been explained at the footnote of the table (Line 256). In addition, the mistake has been corrected according to reviewer suggestion (296).

Round 2

Reviewer 1 Report

The authors have addressed my concerns.

Moderate editing of English language required.